# Recent Advances in Electrochemical Enzyme-Based Biosensors for Food and Beverage Analysis

**DOI:** 10.3390/foods12183355

**Published:** 2023-09-07

**Authors:** Sudarma Dita Wijayanti, Lidiia Tsvik, Dietmar Haltrich

**Affiliations:** 1Laboratory of Food Biotechnology, Department of Food Science and Technology, University of Natural Resources and Life Sciences Vienna, Muthgasse 11, A-1190 Wien, Austria; sudarma.wijayanti@boku.ac.at (S.D.W.);; 2Department of Food Science and Biotechnology, Brawijaya University, Malang 65145, Indonesia

**Keywords:** enzyme, electrochemical biosensor, food analysis, amperometry

## Abstract

Food analysis and control are crucial aspects in food research and production in order to ensure quality and safety of food products. Electrochemical biosensors based on enzymes as the bioreceptors are emerging as promising tools for food analysis because of their high selectivity and sensitivity, short analysis time, and high-cost effectiveness in comparison to conventional methods. This review provides the readers with an overview of various electrochemical enzyme-based biosensors in food analysis, focusing on enzymes used for different applications in the analysis of sugars, alcohols, amino acids and amines, and organic acids, as well as mycotoxins and chemical contaminants. In addition, strategies to improve the performance of enzyme-based biosensors that have been reported over the last five years will be discussed. The challenges and future outlooks for the food sector are also presented.

## 1. Introduction

Food analysis is essential as it provides information on food components, food processing evaluation, as well as food quality, thus ensuring the safety of food, which should meet government and industrial regulations as well as customer satisfaction [1]. Common analytical techniques that are currently being used in food industries include MS (Mass Spectroscopy)-, HPLC (High-performance Liquid Chromatography), NMR (Nuclear Magnetic Resonance)-, PCR (Polymerase Chain Reaction)-, ELISA (Enzyme-linked Immunosorbent Assay)-, and LFD (Lateral Flow Devices)-based methods. These conventional analytical techniques are known for being very labour-intensive, time-consuming, and requiring trained personnel.

A biosensor is an appropriate alternative to the conventional techniques as it provides good selectivity, robustness, high sensitivity, and fast measurements, which are all important requirements for food quality and safety monitoring. A biosensor quantitates biological interaction reactions by producing a signal proportional to the analyte concentration detected. This biorecognition of analytes may involve various bioreceptors including enzymes in combination with a range of transducers generating a signal. The use of enzymes as the bioreceptor in a biosensor was first introduced by Leland C. Clark in 1962, when modifying the ‘Clark electrode’ [2], an electrochemical sensor for oxygen detection, by adding glucose oxidase to the system and thereby creating the first glucose biosensor. Since then, enzymes have been widely employed in the development of biosensors because of their high efficiency and specificity, which enables the detection of low concentrations of analytes together with high selectivity, simplicity, and scalability to industrial levels.

Biosensors for food analysis may be applied in three areas: (i) food safety, detection of contaminants or hazardous substances such as pesticides and bisphenol A, (ii) food quality, which focuses on the determination of compounds that are of interest for nutritional reasons, and (iii) food authenticity, giving information about food origin and verifying that a food product is in compliance with its label description. Electrochemical biosensors are the most common and widely available biosensors currently on the market, as they are portable, user-friendly, and more cost-effective than other types of biosensors. A number of articles on biosensors have been published in recent years as shown in Figure 1, nevertheless, an updated and comprehensive review on the use of enzymes in the development of biosensors related to food applications seems lacking. 

This present review aims to provide information on recent developments of electrochemical enzyme-based biosensors with respect to food analysis over the last five years. Different enzymes used for various applications in food analysis along with the strategies to improve the performance of enzyme-based biosensors will be discussed.

## 2. The Electrochemical Biosensor

A biosensor is an analytical device that integrates a biorecognition element with an electronic component to yield measurable signals [3]. Thus, a biosensor consists of two main components: the biorecognition element and the transducer. The biorecognition element, which is also termed the bioreceptor, is responsible for the specific recognition and interaction with the target analyte [4]. The bioreceptor can be composed of microbial cells, DNA, aptamers, antibodies, or enzymes. The bioreceptor is generally classified as a catalytic or non-catalytic bioreceptor [5], and enzymes, organelles, as well as microorganisms are classified as catalytic bioreceptors. In the catalysis-based biosensors, analytes are reduced or oxidized at the electrode surface by the receptor to produce an analyte-correlated signal. Biorecognition elements such as antibodies and nucleic acids are classified as non-catalytic bioreceptors, which work based on their specific binding affinity toward certain target analytes to trigger a measurable signal.

The transducer is a device that is responsible for the conversion of different types of physical, chemical, or biological reactions into an electrical signal that can then be more easily measured and quantified. Transducers used for biosensors generally depend on the type of material used, the specification of the sensor device, and the mechanism of signal conversion [6]. The signal produced can be designated as optical, thermal, piezoelectric, resulting from a quartz crystal microbalance, and electrochemical. The signal will be further transformed and displayed in a user-friendly way using a signal processor. Table 1 summarizes the working principle, the main benefits, and the disadvantages of different types of transducers potentially used for food application.

Electrochemical biosensors, in particular, are frequently used biosensors and can be categorized as amperometric, potentiometric, conductometric, and impedimetric biosensors. An amperometric biosensor measures either the current or potential resulting from a chemical reaction of electroactive materials on the transducer surface while a constant potential or current, respectively, is applied. When the current is measured at a constant potential, this is referred to as amperometry. If a current is measured during controlled variations in the potential, this is referred to as voltammetry. The concentration of a target analyte is directly proportional to the change in peak current over a linear potential range [7]. Potentiometry measures the potential difference at the working electrode compared to the reference electrode in an electrochemical cell at zero current. Unlike amperometry, the potentiometric response is proportional to the analyte concentration by comparison of its activity to the reference electrode [8]. Conductometry works based on measuring the changes in the sample solution’s conductivity. The interdigitated electrode is the most suitable for conductometric electrodes, which allows the measurement of the conductivity change in the region defined by field lines [9]. Impedimetry measures the changes in charge conductance and capacitance at the sensor surface as the selective binding of the target occurs. Electrochemical impedance spectroscopy (EIS) is a crucial electrochemical method that measures circuit impedance in ohms, a unit of resistance.

## 3. Enzyme Immobilisation

Enzymes are biomolecules with catalytic activity, responsible for increasing biological reaction rates considerably even under mild conditions, and typically show high selectivity. Enzyme-based biosensors function by two possible mechanisms. The enzyme can convert the analyte so that the concentration of the analyte is determined by following its catalytic transformation by the enzyme, or the enzyme can be inhibited by the analyte, so that the concentration of the analyte is associated with a decrease in the formation of the product of the enzymatic reaction [10]. Enzyme catalysis can be affected by several factors including enzyme and substrate concentration, temperature, pH, and the presence of inhibitors or activators. Immobilization or assembly of the enzyme on the electrode surface is a crucial factor in the fabrication of enzyme-based biosensors [11]. Therefore, the selection of appropriate immobilization techniques is essential to provide stability and reproducibility needed for detection. The strategy of immobilization onto the electrode will affect the accessibility of the active site, the stability over time, and the enzyme’s reusability as there is the possibility for the enzyme to become inactivated or leached away from the electrode. Several basic ways to immobilize an enzyme on the electrode have been successfully applied, and these include adsorption, covalent bonding, crosslinking, affinity binding, and entrapment as can be seen in Figure 2A.

Adsorption is a simple technique for the deposition of an enzyme on the electrode surface through weak non-covalent bonds such as Van der Waals forces, hydrophobic interaction, π-π interaction, and/or electrostatic association. This method has been widely used in biosensors because of its versatility [12]. In adsorption, the enzyme is simply deposited by changes in experimental conditions such as temperature, pH, and ionic strength owing to the weak bonding. Covalent attachment offers stable interaction between the enzyme and its support. It prevents the enzyme from leaching from the electrode surface and can thus improve the efficiency of the biosensor [13]. Covalent attachment typically involves certain amino acid side chains that are not essential for the catalytic activity of the enzyme and support the formation of a self-assembled monolayer prior to coupling reactions. These side chains include those of lysine (ε-amino group), cysteine (thiol group), aspartic and glutamic acid (carboxyl group), histidine (imidazole), or tyrosine (phenolic group) [14,15,16]. 

Crosslinking is based on the formation of cross-linkages between the individual enzyme molecules, thus forming a three-dimensional enzyme complex via covalent bonding. Crosslinker reagents commonly used include glutaraldehyde and EDC/NHS [(*N*-ethyl-*N*′-(3-(dimethylamino)propyl carbodiimide/*N*-hydroxysuccinimide)]. This approach provides good stability of the enzyme bound to the surface but may lead to certain losses of activity because of possible severe modifications of the enzymes due to covalent bonding [17,18]. The orientation of a biological molecule that is immobilized on a solid surface is crucial for the development of various applications. The immobilization of biomolecules can use the principle of affinity between complementary molecules such as biotin-avidin. This method benefits from the exceptional selectivity of the interaction. 

Finally, entrapment does not directly attach the enzyme to the surface but encloses it in polymers close to the electrode, which creates a space where substrates and products are freely diffusing in the matrix while the large enzyme is retained. This technique provides high stability and minimization of leaching. Unlike covalent bonding, however, the gel matrix can interfere with the deep diffusion of substrates to the active site of the enzyme. Furthermore, entrapment shows a low loading capacity and positions some of the enzyme molecules far away from the electrode. In some instances, enzyme immobilization protocols are also based on the combination of several immobilization methods since each immobilization method presents different advantages and drawbacks. For example, an enzyme can be pre-immobilized on beads by adsorption or covalent attachment before further being entrapped in a porous polymer [19].

Many studies have compared and modified several basic methods of enzyme immobilisation to determine an optimal method. For example, a functional modification of Au surfaces using thiol-graphene and the addition of HRP-CuP hybrid nanoflowers produced a layer-by-layer self-assembly through Au-S and Cu-S bonds [20]. This strategy improved the detection of pyruvate at microbial fermentation processes by enhancing the catalytic activity and conductivity (Figure 2B). Some conducting polymers such as polyaniline and polypyrrole were also used for the immobilisation of glucose oxidase on the graphite electrode. The polymers were formed by enzymatic polymerization of conducting polymer on the electrode (Figure 2C). The combination of conducting polymers and dendritic gold nanoparticles increased the stability and the detection range of the biosensor [21]. 

**Figure 2 foods-12-03355-f002:**
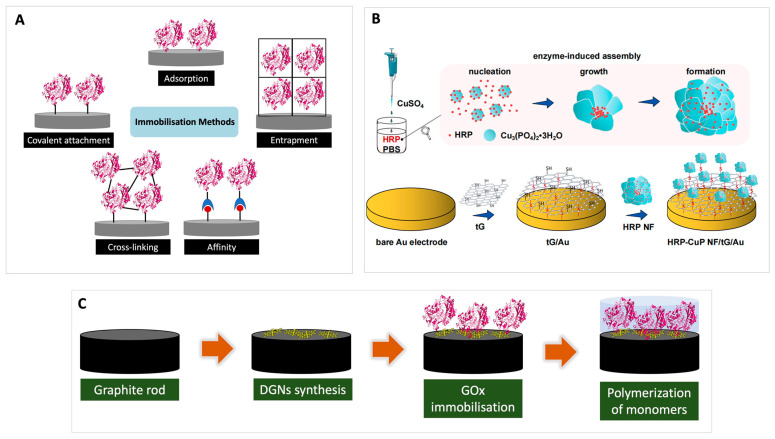
(**A**) Basic enzyme immobilisation methods in biosensors. Modifications of immobilisation methods in an enzyme biosensor for (**B**) pyruvate detection (reprinted with permission from Ref. [20], Copyright 2023, Elsevier), and (**C**) glucose detection.

## 4. Enzyme-Based Biosensors for Food Analysis

### 4.1. Saccharides

Saccharides or carbohydrates are widely distributed in foods and beverages, and in addition to their natural occurrence they are often used as food additives as well. The classification of saccharides is based on their degree of polymerization, dividing them into three principal groups, i.e., sugars, oligosaccharides, and polysaccharides. Sugars with a degree of polymerization (DP) of 1–2 comprise monosaccharides, disaccharides, and polyols, oligosaccharides with their DP of 3–9 include, for example, malto-oligosaccharides, and polysaccharides show a DP of 9 or more [22,23]. The monitoring of saccharides in food is important since it provides information for nutrition labelling, quality assurance, and food sensory evaluation.

#### 4.1.1. Glucose

Glucose clearly dominates among saccharides when reported as analyte for electrochemical enzyme biosensors. Glucose biosensors are frequently applied in the dairy, wine, beer, and sugar industries as well as for monitoring various fermented products. Determination of the glucose content can be crucial for some foods since glucose causes browning during dehydration and processing, mainly due to the Maillard reaction. Enzymatic amperometric glucose biosensors are the most prevalent devices commercially available, and have been extensively studied over the last few decades. Most amperometric glucose biosensors are based on glucose oxidase (GOx, EC 1.1.3.4), which employs oxygen as its electron acceptor to selectively catalyse the oxidation of β-D-glucose to β-D-glucono-1,5-lactone and hydrogen peroxide [24]. For these applications, GOx is mainly obtained from fungal sources such as *Aspergillus niger* (*An*GOx) [21,25,26,27,28] and *Penicillium vitale* [29,30]. *An*GOx is especially well known and preferred because of its stability over a wide range of temperature and pH together with its excellent selectivity for glucose. However, since GOx utilizes oxygen as the electron acceptor, the biosensor will be sensitive to oxygen concentrations and may show fluctuations and errors in measurements due to variations in O_2_ concentrations. In order to overcome this challenge and improve sensing reliability, glucose dehydrogenase (GDH), which is not dependent on oxygen, has been studied as an alternative for various sensing applications [31,32,33]. Despite this advantage, commercial GDH-based biosensors have not yet been developed widely because of its lack of specificity for glucose and low reactivity with other sugars such as maltose in comparison to GOx. Another enzyme of fungal origin, pyranose oxidase (POx, EC 1.1.3.10) from *Trametes multicolor*, has been tested for biosensor applications as well [34]. The POx-based biosensor not only detects glucose but also galactose, and shows the advantage of reacting with both anomeric forms of glucose. The majority of glucose biosensors reported to date are of the first- and second-generation type. First-generation glucose biosensors directly measure hydrogen peroxide, which is oxidised by an applied voltage at the electrode surface, thus producing an electric signal. This can have the disadvantage of applying high overpotentials and electrode poisoning. To overcome these drawbacks, second-generation biosensors were introduced, which use redox mediators with a lower redox potential that shuttle electrons between the prosthetic group and the electrode. However, second-generation biosensors also result in a more complicated sensor architecture, and leaking of the mediators maybe occur. Jayakumar and co-workers reported the use of the enzyme cellobiose dehydrogenase (CDH) from *Crassicarpon hotsonii*, which was engineered by incorporating the mutations C291Y and W295R to enhance its activity with glucose. The sensor set-up enabled both mediated (MET) and direct electron transfer (DET) to the electrode. In DET, which forms the basis of third-generation biosensors, the electrons are directly transferred from the haem group of the enzyme to the electrode. The CDH-based sensor offered a substantial improvement of sensitivity over other DET-based glucose biosensors and showed no dependency on oxygen [35]. Even though the application of this third-generation glucose biosensor was mainly aiming at use in clinical applications, it shows promise for food applications in the near future. The use of third-generation biosensors for glucose detection in real food samples has not been reported to date to the best of our knowledge. Table 2 summarizes the analytical performance of glucose biosensors in food and beverages samples. 

#### 4.1.2. Other Saccharides

Fructose is a naturally occurring monosaccharide widely found in fruits and in lesser amounts in tuberous vegetables such as onions and potatoes [40]. Detection of fructose is important as fructose has been increasingly used in the food industry because of its sweetness and therefore is added as a sweetener. Fructose dehydrogenase (FDH, EC 1.1.99.11) from *Gluconobacter japonicus* has been widely studied for the development of biosensors based on MET and DET [41,42]. Fructose dehydrogenase catalyses the oxidation of D-fructose to 5-keto-D-fructose in the presence of suitable electron acceptors. The ability of FDH to conduct DET makes it a promising enzyme for food applications in view of the toxicity of some mediators used in MET. DET provides simplicity in reactions and is less prone to interfering reactions offering good selectivity. A fructose biosensor based on FDH was reported [43], which showed an efficient DET reaction pathway between the enzyme and a glassy carbon electrode (GCE) modified with single-walled carbon nanotubes (SWCNTs) and through diazonium coupling of an aromatic compound. The electrode modification was carried out to allow an interaction between the aromatic anthracenyl groups available on the electrode surface and the hydrophobic region within the enzyme. The proposed fructose biosensor resulted in a high sensitivity of 47 μA·mM^−1^·cm^−2^ and a low limit of detection (LOD) of 0.9 μM when compared to the biosensor without electrodepositing anthracene onto SWCNT.

In contrast to monosaccharide sensing, which commonly involves a single enzymatic reaction, a cascade enzyme system is typically used as the biorecognition element for the detection of disaccharides. Sucrose biosensors, for example, use invertase to hydrolyse sucrose to fructose and α-D-glucose, then α-D-glucose is converted to β-D-glucose by mutarotase, and finally β-D-glucose is oxidized/detected by either GOx or GDH, which both are specific for this anomer of glucose. A bi-enzymatic reaction involving invertase and GOx was investigated using microfibrillated cellulose to form a nanobiocomposite-based sucrose biosensor using screen-printed gold electrodes [44]. The biosensor showed a wide range of detection from 0.1 nM to 10 μM with storage stability up to 4 months. Stredansky and co-workers developed a sucrose biosensor based on the three enzymes invertase, mutarotase, and GDH. Instead of using complex layers for the fabrication of the biosensor, they applied a simple and effective co-immobilisation based on chitosan layers on the surface of thin-layer planar gold electrodes. The platform showed a low LOD and excellent storage stability, retaining more than 90% of the initial response value after 12 months of storage [45].

Bi-enzymatic biosensors are often used in the analysis of disaccharides, particularly for maltose detection. The measurement of the maltose concentration is especially important in the brewing industry as it may influence the sensory characteristics of the final product. Typically, two enzymes, α-1,4-glucosidase and GOx, are used to this end [46,47,48,49]. A recent report on a maltose biosensor developed by our group shows the possibility of a single enzyme capable of maltose detection [50]. GDH from *Trichoderma virens* from the until then unexplored GDH class II was used as the biorecognition element and wired by an osmium redox polymer to a graphite electrode. The biosensor showed an LOD of 0.45 mM towards maltose and could also detect glucose, maltotriose, and galactose.

Lactose biosensors are of growing interest among biosensors for saccharide detection. The increase in lactose intolerance among customers and its awareness resulted in a market preference shifting towards lactose-free products. Thus, it is important to have a sensitive, rapid, and accurate method available for the detection of very low levels of lactose in food products, and especially in dairy products. Enzyme-based biosensors for lactose detection in food typically reported the application of a multienzyme cascade reaction. β-Galactosidase or lactase (β-gal, EC 3.2.1.23) from *Aspergillus oryzae* [51,52,53] has been predominantly used for the first step of the cascade, the hydrolysis of lactose. The enzyme is extracellularly formed by this fungus, it is thermostable, shows high activity, and carries the GRAS status [54]. Galactose can then be analysed/oxidized by either galactose oxidase (GalOx, EC 1.1.3.9) from *Dactylium dendroides* [51] or *Aspergillus niger* [52]. Alternatively, glucose can be analysed by glucose oxidase [55,56]. Cutting-edge biosensors known as photoelectrochemical (PEC) biosensors have recently been developed. These systems offer zero-potential detection feasibility and are highly sought after due to their ability to maintain enzyme bioactivity, reduced interference effects, and intrinsic sensitivity. A novel PEC multi-analyte biosensor was developed [55] that used semiconductor-metal NPs containing nanomaterial for the effective shuttling of electron to the electrode surface (Figure 3). The detection produced a good linear measurement range, and sensitivity for both glucose and lactose as analytes [55].

A single enzyme, cellobiose dehydrogenase (CDH), had been investigated as an alternative to these two-enzyme systems for the development of a third-generation lactose biosensor. A number of publications pointed out the potential of CDH, particularly from CDH class I and II, as biorecognition element for lactose biosensors [57,58,59]. The commercially available sensor system Lactosens was first launched by the company DirectSens in 2017. Lactosens is the only third-generation biosensor on the market that is able to detect very low lactose concentrations for applications in dairy companies (www.lactosens.com (accessed on 13 April 2023)). The potential application of enzyme-based biosensor in saccharide detection is summarised in Table 3.

### 4.2. Alcohol Beverages—Ethanol and Antioxidants 

The detection of alcohol (ethanol) is an important feature in assuring the quality of fermented food products, particularly for wine and beer. The ethanol content not only provides information on the progress and the optimization of the fermentation process but also plays roles in stability, ageing, and the sensory properties of the fermented products. As an example, knowledge of the total alcohol content in wine is necessary for grading. Wine usually contains between 9–15% (*v*/*v*) ethanol, which is higher than in most other fermented beverages because of the high concentration of sugar in grapes [64,65]. An amount of alcohol less than that or exceeding 17.5% (*v*/*v*) is typically an indication of wine adulteration. For the purpose of ensuring beverages are Halal compliant and certified, quantitating very low levels of ethanol is also important. In order to be Halal compliant and certified, foods must contain less than 1% ethanol that had been formed by a natural (aerobic) fermentation process. However, this level is lower for beverages, and they must contain less than 0.1% ethanol to be classified as Halal [65]. 

Alcohol oxidase (AOx, EC 1.1.3.13) and alcohol dehydrogenase (ADH, EC 1.1.1.1) from yeast, and here especially from *Saccharomyces cerevisiae*, are commonly used enzymes for alcohol biosensor applications. AOx catalyses the oxidation of alcohols into the corresponding aldehydes or ketones, but not the reverse reaction as is the case by ADH. In addition, the two enzymes show differences in their cofactor requirement. AOx contains flavin-based cofactors, while ADH requires NAD-based coenzymes [66]. ADH is a widely used enzyme in the development of enzymatic biosensors for detecting ethanol. The detection of the substrate is accurately achieved by measuring the quantity of NADH produced during the enzymatic reaction. A study reported the utilization of ADH entrapped into a sol-gel matrix that was immobilized on the surface of a screen-printed electrode modified with poly(allylamine hydrochloride) [67]. Amperometry result showed a low LOD of 20 μM and a wide dynamic response range. The measurement of ethanol in actual beer samples showed a good recovery in agreement with the ethanol content declared by the manufacturer [67]. Conductometric biosensors, first reported to detect gaseous ethanol, showed not only a good response time compared to ones using an amperometric transducer but also showed higher detections limit [68] as can be seen in Figure 4.

Polyphenols are commonly found in beer and wine products. Polyphenols in red wines, for example, are a complex mixture of flavonoids, which act as potent antioxidants contributing to health benefits, especially on the cardiovascular system [69]. Laccase (Lacc, EC 1.10.3.2) or tyrosinase (Tyr, EC 1.14.18.1) have been frequently used in biosensor applications due to their respective substrates, *o*-diphenols or monophenols, which are also related to the antioxidant capacity of food and beverages. Bellido-Milla and co-workers developed a new electrodeposition method based on the use of sinusoidal current electrodeposition to immobilize tyrosinase onto sonogel-carbon electrodes, generating a nanostructured surface for the fabrication of a polyphenol biosensor. The biosensor displayed good analytical performance, and it had been applied to determine the polyphenol index in commercially available beer and wine samples [70]. A biosensor based on a nanostructured, functional platform involving laccase immobilized on a AuNPs/Screen-Printed Carbon Electrode (SPCE) modified with polypyrrole was synthesized and showed a low detection limit when applied in propolis [71]. Relatively few studies focusing on enzyme-based biosensors for antioxidant detection have been reported in the past 5 year though, as DNA-based biosensor approaches are nowadays preferably used to evaluate the antioxidant capacity in food. These DNA-based biosensors use DNA as the recognition element, and their performance is assessed based on the principle of oxidative damage inflicted on the DNA by radicals such as reactive oxygen species. By adding an antioxidant to the solution, the oxidative damage should decrease, indirectly evaluating the antioxidant capacity of the sample. This approach is currently preferred as it closer to what occurs in cells [72]. 

Table 4 summarises the analytical performances of various biosensors when applied for the detection of alcohol and antioxidants in beverages.

### 4.3. Organic Acids 

Organic acids play a number of roles in the food and beverage industry. A recent review on organic acids in food emphasised that these compounds benefit both nutrition and human health considerably. Moderate amounts of consumed acids were proven to regulate the metabolism and provide energy while safeguarding the immune and myocardial systems [78]. Moreover, organic acids significantly contribute to the flavours and aroma by enhancing and inhibiting other taste sensations and, more prominently, building up an inherent taste of consumed food. The impact of organic acids extends beyond flavour perception alone, as their presence and concentration in food affect various aspects of quality, safety, and preservation. Monitoring fermentation processes and assessing the freshness of processed food material is part of the critical control points in food technology, where enzyme-based biosensors are mainly applied. 

Lactic acid is a universal organic acid that when present reflects several aspects of food and beverages, including quality, stability, and organoleptic characteristics. Several recent studies [79,80,81,82] indicate that the dairy and wine industries would benefit significantly if the evaluation of lactate level can be performed with more sensitive lactate biosensors. For instance, a highly sensitive biosensor for L-lactate determination based on direct electron transfer enzymes was developed [81]. These authors improved existing biosensors by enhancing the electron transfer between the electrode surface and the enzyme cytochrome c oxidoreductase (flavocytochrome *b*_2_, EC 1.1.2.3) by applying bimetallic and trimetallic nanoparticles, and verified the best electrode set-up with yoghurt samples. Briefly, PtZn, NiPtPd, and hexacyanoferrates of Au (AuHCF) served as active redox mediators and were safely co-immobilized with L-lactic specific cytochrome c oxidoreductase. A modification of the working electrode surface with AuHCF led to a 3.5-fold increased sensitivity and a 2.7-fold decreased LOD compared to a control biosensor composed of bare L-lactate-cytochrome c oxidoreductase immobilised on the graphite electrode [81]. Ozoglu and co-workers studied the detection of lactate produced by lactic acid bacteria isolated from cheese samples. The authors pointed out that applying biosensors for the detection of bacterial metabolites without pre-treatment can be challenging due to the presence of other organic acids, which also may be oxidised at the working electrode [82]. This is especially a problem when dealing with biosensor architectures that require higher potentials for current production, such as depletion of hydrogen peroxide, as observed in Ozoglu’s study. A possible approach to mitigate the problem of current interferences from fermentation metabolites and high potentials is the use of compound-specific mediated systems, where direct interaction of hydrogen peroxide with the working electrode is avoided. Zhou and co-workers outlined this approach for the effective detection of pyruvic acid produced during yeast fermentations [20]. As an important intermediate of the microbial metabolism, pyruvic acid detection is key to controlling fermentation processes in the food industry. The reported biosensor was based on a system applying decarboxylation of pyruvic acid after the reaction with hydrogen peroxide. Horseradish peroxidase (EC 1.11.1.7) as the main biocatalyst was regenerated by co-immobilized Cu^+^, ensuring effective readout of the hydrogen peroxide reduction proportional to the decrease in pyruvic acid levels in the metabolite sample [20]. This approach presented an excellent alternative for earlier reported biosensors [83] that used the principle of direct interaction between potassium ferricyanide (mediator) and hydrogen peroxide. 

Over the last five years, no significant increase in reports regarding new methods of organic acid detection was made. Many of the biosensors used for detecting food-related organic acids have undergone slight modifications to their existing architecture. These modifications were mainly focusing on incorporating novel nanocomposites or electroactive particles to increase sensor sensitivity (see Table 5). A big spike in publications was noted for lactate-specific biosensors explicitly developed for medical and patient care purposes though. This work is intentionally not covered in the current review, as the application of these sensors has not been tested in food samples or for the food industry. However, the currently very active research on medical lactate sensors may simplify or accelerate the development of multi-analyte biosensors for complex food matrices. Another study announced a sophisticated enzyme-based sensor system for multiparametric detection of three organic acids and ethanol [84]. The sensor design included four enzymes, *Candida boidini* formate dehydrogenase (EC 1.2.1.2), *Bacillus stearothermophilus* and *Lactobacillus leichmanii* lactate dehydrogenases (specific for the L-lactate isoform, EC 1.1.1.27, and the D-lactate isomer, EC 1.1.1.28, respectively), as well as *Saccharomyces cerevisiae* alcohol dehydrogenase (EC.1.1.1.1). The detection principle was similar for these four biocatalysts. It relied on the anodic oxidation of the enzymatically produced mediator K_4_[Fe(CN)_6_] in the presence of the regenerating cofactor NAD^+^ (Figure 5). 

Notably, this biosensor could distinguish between the isoforms of lactic acid, which is extremely valuable for malolactic and alcohol fermentation processes in beverage and food production [84]. Later, this biosensor design was modified further to enable monitoring of volatile short-chain fatty acids as well. For instance, acetate and propionate are known to balance anaerobic digestion in biogas production, which is carried out to manage food waste. The authors reported the integration of *Clostridium propionicum* propionate CoA-transferase (EC 2.8.3.1) and short-chain *Arabidopsis thaliana* acyl-CoA oxidase (EC 1.3.3.6) for the electrochemical sensing of propionate. Furthermore, immobilisation of *Escherichia coli* acetate kinase, pyruvate kinase from rabbit muscle, and *Aerococcus viridans* pyruvate oxidase was used to indirectly detect acetic acid levels [86]. Although the multi-analyte biosensor was reported in this study, the synchronous measurement of six organic acids was not achieved. The volatile fatty acid biosensor operated at a distinct working potential that did not match the detection principle of the organic acid biosensor developed by [84]. A year later, another attempt to design a multi-analyte sensor was reported [87]. Interestingly, the main focus of this study was to deliver a novel method for wine classification based on trained artificial neural networks. A multipurpose sensor was needed to collect comprehensive information regarding the carboxylic acid content of 31 wine samples. Specifically, a two-channel biosensor was developed that was composed of lactate oxidase and sarcosine oxidase (SOx, EC1.5.3.1) co-immobilized with fumarase (FUM, EC 4.2.1.2). While the lactate oxidase-based sensor ensured anodic oxidation of enzymatically generated hydrogen peroxide, the second channel, composed of two enzymes, recorded the inhibitory effect of carboxylic acids from wine on the conversion of sarcosine with SOx. The presence of FUM ensured the enzymatic conversion of tartaric acid, which could inhibit SOx [87]. The concepts of multipurpose biosensors enlarge the possibilities for organic acid detection in complex food matrices. Moreover, it highlights that different enzymes can be immobilised in a unified sensing system, and different detection methods can be combined in a single sensing device. 

**Table 5 foods-12-03355-t005:** Application of enzyme-based biosensor for organic acids detection in food samples.

Analyte	Electrode	Enzyme	Transducer	Sensitivity	Detection Range	LOD	Food Matrices	Ref.
L-Lactate	Pt/Ti/GA/BSA/ Glycerol	L-LDH, DIA	Amp	37.2 μA mM^−1^ cm^−2^	n.d	0.7 μM	Maize and sugarcane silage	[84]
D-lactate		D-LDH, DIA		28.4 μA mM^−1^ cm^−2^		0.7 μM		
Formate		FDH, DIA		20.5 μA mM^−1^ cm^−2^		1.3 μM		
Acetate	Pt/GA/BSA/ Glycerol	AK, PK, PyOx	Amp	0.27 µA mM^−1^	0–1.4 mM	n.d	Food waste	[86]
Propionate		PCT, SCAOx		2.11 µA mM^−1^	0–1.5 mM			
L-Lactate	Cu-MOF/CS/Pt/SPCE	LOx	Amp	14.65 µA mM^−1^; under inhibition 0.207 µA mM^−1^	0.00075–1.0 mM; under inhibition 4.0–50 mM	0.75 μM	Red and white wines	[88]
L-Lactate	Pt/rGO/CNT/Au	LOx	Amp	35.3 μA mM^−1^ cm^−2^	0.05–100 mM	2.3 μM	Cow milk	[89]
L-Lactate	Pt/OPD/resorcinol/GA/BSA	LOx	Amp	n.d	0.05–4.5 mM	0.03 mM	Red and white wines	[87]
Malic, tartaric acids		SOx, FUM						
L-Lactate	Pt/Pd/BSA/GA/ Dextran/Lactitol/Glycerol	LOx	Amp	3.03 μA mM^−1^ cm^−2^	0.05–0.8 mM	0.1 µM	Red and white wines	[90]
L-Lactate	GA/AuNPs-ERGO-PAH/SPE	L-LDH	Amp	I range: 1.08 μA mM^−1^cm^−2^; II range: 0.28 μA mM^−1^cm^−2^	I range: 0.5–3 mM; II range: 4–16 mM	1 µM	Yoghurt and wine	[79]
L-Lactate	Aluminium coated cellulose	LOx	Amp	10.04 μA mM^−1^ cm^−2^	0.125–2 M	0.23 M	Cow milk	[91]
L-Lactate	CF-H/PtMPs	LOx	Amp	5233 A M^−1^m^−2^	0.005 mM–0.14 mM	2 μM	Red wine	[80]
L-Lactate	PtZn/GE + PMS	Fcb2	Amp	1436 A M^−1^m^−2^	0.01–0.12 mM	0.01 mM	Yoghurt	[81]
L-Lactate	Pt/Nafion	LOx	Amp	0.4 µA mM^−1^ cm²	50–350 µM	31 µM	Lactic acid bacteria metabolites	[82]
Pyruvate	Cu-NF/tG/Au	HRP	Amp	67.6 μA mM^−1^ cm^−2^	0.1–8.2 mM	0.06 mM	Yeast metabolites	[20]
Pyruvate	GQD/PB/SPCE	POx	Amp	40.8 μA mM^−1^ cm^−2^	10–100 μM	0.91 μM	Fish serum samples	[83]

LOD, limit of detection; FDH, formate dehydrogenase; L-LDH, L-lactate dehydrogenase; D-LDH, D-lactate dehydrogenase; GA, glutaraldehyde; BSA, bovine serum albumin; DIA, diaphorase; Cu-MOF, copper metallic framework; Pt (MPs), platinum (microparticles); CS, chitosan; SPCE, screen-printed carbon electrode; SPE, screen printed electrodes; CNT, carbon nanotubes; rGO (or ERGO), reduced graphene oxide; Au (AuNps), gold (nanoparticles); OPD, o-Phenylenediamine; PAH, poly(allylamine hydrochloride); CF-H, hemin-functionalised carbon microfibers; GE, graphite rod; PMS, phenazine methosulfate; Cu-NF, Cu_3_(PO_4_)_2_·3H_2_O nanoflowers; tG, thiol graphene; GQD, graphene quantum dot; PB, prussian blue nanoparticles; LOx, lactate oxidase; FUM, fumarase; Fcb2, L-lactate-cytochrome c oxidoreductase; PyOx, pyruvate oxidase; PCT, propionate CoA-transferase; SCAOx, short chain acyl-CoA oxidase; PK, pyruvate kinase; AK, acetate kinase; Amp, amperometry; n.d, not determined.

### 4.4. Amino Acids, Biogenic Amines, and Purine Derivatives 

Protein quality in food products is usually assessed through an analysis of the amino acid composition of the protein and hence it is important that the concentration of amino acids (especially essential amino acids) can be accurately determined. L-Lysine is an essential amino acid commonly found in protein-rich food such as meat and cheese. A recent study developed a potentiometric lysine biosensor based on lysine oxidase (LyOx, EC 1.4.3.14) from *Pediococcus* sp. in combination with an oxygen electrode [92]. Optimization studies were carried out to measure the lysine content in mozzarella cheese, an important parameter to evaluate the maturity of the cheese. The LOD of this biosensor was higher when compared to an amperometric lysine biosensor using nanoparticles of LyOx from *Trichoderma viride* [93], yet it also showed a wider detection range. L-Glutamate is a non-essential amino acid, which occurs naturally in a range of foods including tomatoes, cheese, and mushrooms, and plays an important role in the palatability of food. L-Glutamate oxidase (GluOx, EC 1.4.3.11) catalyses the oxidative deamination of the α-amino group of L-glutamate to 2-ketoglutarate, with concomitant reduction of molecular oxygen and water to ammonia and hydrogen peroxide. Monitoring glutamate concentrations can then be performed by measuring the hydrogen peroxide concentrations or the oxygen consumption. A recent study constructed a glutamate sensor for detecting glutamate in different growth stages of tomatoes. The biosensor prototype was reported to be practical by monitoring the glutamate level in situ by inserting the sensor directly into the tomato. The authors also claimed that the biosensor performed the widest detection range to date of 2 μM–16 mM [94]. This range is a suitable range for food application, for example, the glutamate content in watermelon is ~12 mM and around 10 mM in tomatoes. A tailored GluOx immobilization procedure by co-crosslinking the enzyme with bovine serum albumin (BSA) and glutaraldehyde onto a polymer-modified electrode was developed. Thereby, a high sensitivity of 18.3 mA·M^−1^cm^−2^ was obtained. Additionally, the disposable biosensor resulted in accurate glutamate analyses in complex food matrices with a good sample throughput [95].

Biogenic amines (BAs) are low molecular mass nitrogen compounds with biological activity present in microorganisms, plants, and animals, with some important BAs including histamine, tyramine, tryptamine, putrescine, and cadaverine. BAs are formed and degraded as part of the normal metabolism of organisms [96]. Among the BAs, histamine is found to be the most important BA with respect to food spoilage. Hence, detection of histamine levels in food is an important aspect of food safety as histamine is one of the main causative agents for food intolerance and poisoning. The presence of excess concentrations of histamine is usually considered as an indicator of decomposition and bacterial spoilage or activity. In the last years, various electrochemical sensors for BA detection were developed based on screen-printed electrodes, which is user-friendly and offers the additional advantage of on-site analysis, utilizing either diamine oxidase (DAO, EC 1.4.3.6) from porcine kidney or monoamine oxidase (MAO, EC 1.4.3.4). Both amine oxidases catalyse the oxidative deamination of histamine to imidazole acetaldehyde, hydrogen peroxide and ammonia. Koçoǧlu and co-workers reported two different biosensors using a single enzyme, either DAO or MAO, as the biorecognition element immobilized on the SPCE modified with a mixture of titanium dioxide nanoparticles (TiO_2_), carboxylated multi-walled carbon nanotubes (MWCNT), hexaammineruthenium(III) chloride, and chitosan. The analytical performance comparison of the two biosensors showed that the sensor based on DAO had a wider linear range and 1.5-times higher sensitivity than the MAO-based biosensor [97]. This was also verified by another study showing the construction of a histamine biosensor using a metal oxide nanoparticle-Prussian Blue-modified electrode showing that a sensor using DAO gave a 30-fold higher sensitivity compared to MAO [98]. Different fabrication methods and materials used in the immobilization of the enzyme also affect the performance of a biosensor. A tyramine biosensor using MAO showed 1.5-fold higher sensitivity than one using DAO, while the LOD and the response time showed no significant difference [99].

The use of bi-enzymatic biosensors combining diamine oxidase with horseradish peroxidase (HRP) was reported for the detection of BAs as well. Both enzymes were immobilized onto the surface of the SPCE using glutaraldehyde (GA) and BSA cross-linking. The biosensor was fabricated for the detection of histamine-producing bacteria by measuring the signal generated in the presence of histamine, producing a good analytical performance [100]. The same crosslinking method using GA and BSA was also used by [101] to immobilize DAO. They found that the concentration of GA had a more pronounced effect on the response of the sensor than BSA. BSA and GA concentrations of 3% and 0.5% were selected, which was considered as the best compromise between the analytical signal and the precision of the biosensor [101]. In a later study, they simplified the biosensor construction by only using GA for the immobilization of DAO and optimisation of the electrochemical oxidation by adding ferricyanide ([Fe(CN)6]^3−^) as a redox mediator, thereby improving the sensitivity more than 100-fold [102]. Hidouri and co-workers reported a sensitive bi-enzymatic, potentiometric histamine sensor, which produced a highly specific response to histamine and showed the lowest detection limit of less than 10 nM [103]. The use of DAO commonly generates hydrogen peroxidase (H_2_O_2_) which requires the application of a high potential when it is to be detected by oxidation on the electrode. This in turn can create interfering signals from other electroactive species. Incorporating Prussian blue [iron(III) ferrocyanide], which acts as an artificial peroxidase, to modify the electrode enables H_2_O_2_ detection at a lower potential and thus can avoid interference [104]. A smart electrochemical biosensor based on the DAO-PANI/ZnO@TiO_2_@n-C_22_ MEPCM-modified GCE as a working electrode was developed as illustrated in Figure 6. The strategy aimed for enhancing histamine detection in high-temperature environments. The biosensor showed higher sensitivity, and lower LOD at a high assay temperature compared to conventional biosensors without phase charge materials [105]. 

Hypoxanthine and xanthine are intermediate compounds formed in the degradation of purines to uric acid. These two molecules are interesting biomarkers for the freshness of fish. After the death of a fish, ATP gets degraded into xanthine, and its concentration increases during storage with time. The biorecognition element used in the majority of electrochemical biosensors for xanthine detection is a xanthine oxidase (XOx, EC 1.17.3.2), which catalyses the oxidation of hypoxanthine to xanthine and then xanthine to uric acid with concomitant reduction of molecular oxygen. Immobilization of both XOx and uricase (EC 1.7.3.3) on polypyrrole-paratoluenesulfonate via entrapment was reported by Erol and co-workers for the detection of hypoxanthine. Uricase catalyses the oxidation of uric acid into 5-hydroxyisourate while reducing dioxygen into hydrogen peroxide. Hypoxanthine analysis was performed based on the 0.3 V oxidation of hydrogen peroxide formed as a result of enzymatic reaction sequences on the surface of the electrode [106]. Since XOx catalyses both the oxidation of hypoxanthine to xanthine and xanthine to uric acid, XOx biosensors based on uric acid as well as hydrogen peroxide or oxygen consumption measurements indicate the total concentration of hypoxanthine and xanthine present in samples. Armada and co-workers reported that optimum conditions for the anodic measurements are at a working potential of 0.4 V when measuring the hydrogen peroxide production as a result of the enzymatic reactions, resulting in two wide linear sections (0.03–0.2 and 0.2–0.8 mM xanthine) and a competitive LOD (30 nM). A novel aspect of this xanthine biosensor was that when performing the measurement of xanthine through the enzymatic consumption of oxygen at a potential −0.1 V, an improved sensitivity was observed [107]. Another study reported an even lower detection limit of 1.14 nM xanthine by covalent immobilization of XOx from *Bacillus pumilus* RL-2d onto a screen-printed multi-walled carbon nanotubes gold nanoparticle-based electrode (Nano-Au/c-MWCNT). The biosensor was able to detect fish freshness by comparing xanthine levels from fresh fish and 5-day old fish sample [108]. A summary of analytical performances of various biosensors for the detection of amino acids, biogenic amines and purine derivative compounds is given in Table 6.

### 4.5. Chemical Contaminants

#### 4.5.1. Pesticides

Organophosphorus pesticides (OPs) are persistent chemical compounds preventing the proliferation of various pests, and are widely used in agriculture as plant protection agents in order to improve the quality and quantity of crops. However, OPs are toxic to humans and most animals, causing damage to the nervous system. The construction of electrochemical biosensors for OP detection has mainly been based on the ability of OPs to inhibit acetylcholinesterase. Acetylcholinesterase (AChE, EC 3.1.1.7) is a key enzyme for the proper functioning of the central nervous system in both humans and insects. Interaction of acetylthiocholine chloride (ATCl) with AChE will produce thiocholine (TCl) [110]. The working principle of AChE biosensors depends on the high affinity of OPs towards AChE, which causes a decrease in ATCl hydrolysis, thus the decrease in the electrochemical signal output.

Jiang and co-workers fabricated a novel OP biosensor, particularly for malathion and methyl parathion detection based on electrostatic self-assembly in combination with in situ photo-cross-linking of AChE on Prussian blue (PB) deposited on the single-walled carbon nanotube (PB-SWCNTs) backbone. The biosensor showed a lower LOD for malathion than for methyl parathion [111]. Palanivelu and Chidambaram reported the utilization of mesoporous silica nanoparticles, Santa Barbara Amorphous (SBA-15), as a carrier to cap (immobilize) AChE, and thereby developed a sensitive and particularly stable biorecognition complex for the biosensor. They tested and validated it for its LODs of various pesticides in soft drinks, comparing it to Ellman’s method, the routine method to measure cholinesterase activities. The minimal detection level for monocrotophos and dimethoate were 2.5 and 1.5 ppb, respectively, and the sensor was regarded to be thermally stable when compared with other biosensors [112].

An investigation of the effect of doping Au nanorod core shell nanoparticles formed on mesoporous silica (AuNRs@MS) into the TiO_2_-chitosan film hydrogel was conducted in relation to an OP biosensor [113]. The hydrogel films have a mesoporous nanostructure, which in a previous study showed an improved AChE loading efficiency and thus produced a stable AChE biosensor [114]. The MS shell was thought to protect the AuNRs from aggregation, while its porous structure should allow the transport of ions, ensuring the enhancement effect of AuNRs on the electroconductivity and electrocatalytic activity. In addition, small molecules are able to permeate through the MS shell and assure contact of TCl with the electrocatalytic AuNRs. The biosensor was used to detect dichlorvos and fenthion (Figure 7). The doping significantly enhanced the electroconductivity of the TiO_2_-chitosan hydrogel and dramatically improved the bioelectrocatalytic activity and OP detection sensitivity of the immobilized AChE matrix. 

Another study also reported a biosensor for the detection of dichlorvos with a patterned structure of AChE-CS/TiO_2_-CS on glassy carbon electrodes. The pattern structure helped to avoid non-conductive substances such as enzymes and chitosan blocking the electron transmission and mitigates the current loss of the sensor. The biosensor produced a lower LOD for dichlorvos [115].

Most of the AChE used for the biorecognition element of the biosensor originates from the electric eel *Electrophorus electricus* [112,113,115,116]. Hayat and co-workers investigated the potential of crude AChE preparations from *Tribolium castaneum* (red flour beetle) for an electrochemical biosensor for detection of the organophosphate insecticide, phosmet. A novel WO_3_/g-C_3_N_4_ nanocomposite material, made by doping the graphitic carbon nitride (g-C_3_N_4_) nanomaterial with tungsten trioxide (WO_3_), was used to modify the pencil graphite electrode to improve the electron transfer between the enzyme and electrode surface, which in turn improved the electrical conductivity and quality of the output signal with minimal interfering effects. The results showed that AChE activity predominantly increased when incorporated within the nanocomposite. When measuring phosmet in whole wheat flour, the percentage recoveries were found to be up to 99%, and obtained results were in agreement with the results of HPLC analysis [117].

The majority of biosensors for the detection of organophosphorus pesticides typically detect only few compounds. Zhao and co-workers used an indirect competition method. They constructed a nanogold/mercaptomethamidophos multi-residue electrochemical biosensor utilizing a combination of nanotechnology, surface chemical modification, and biosensor technology to simultaneously detect eleven OPs, and applied this sensor to the detection of OPs in realistic samples of apple and cabbage. The biosensor produced has a good correlation coefficient [118]. 

A simple MOF (Metal Organic Framework)-based immobilization-free electrochemical sensing strategy for the detection of pesticide residues was developed [119]. Using this approach, they achieved excellent analytical performance for the detection of paraoxon with a detection limit of 1.7 ng·mL^−1^, and the biosensor was claimed to allow the simultaneous determination of not only OPs but also carbamates [119]. This was also verified by Bagheri’s group who developed MOF-based sensing for the detection of paraoxon. The addition of Ce in a Zr-based MOF structure in combination with MWCNTs demonstrated a rapid and sensitive detection of paraoxon with a slightly higher LOD [116].

#### 4.5.2. Bisphenol A (BPA)

Bisphenol A or BPA is a chemical compound primarily used in the manufacturing of various packaging systems (plastics such as polycarbonates and resins). BPA can leach into food or beverages from BPA-containing containers, which leads to accumulation of BPA in the human body and thus can causing health problems. Exposure to BPA is a concern because of its possible effects on the brain and the prostate gland. Tyrosinase (Tyr, EC 1.14.18), belonging to the group of phenol oxidases, is found widespread in nature and catalyses the oxidation of a wide range of phenolic compounds including BPA. Moreover, Tyr has a higher specificity toward BPA than other polyphenol oxidases.

A layer-by-layer assembly of Tyr in an ultrathin copper-porphyrin MOF nanofilm (Tyr@Cu–TCPP) via a simple one-step solvothermal method. Based on this nanofilm was developed [120]. They fabricated an ultrasensitive electrochemical biosensor for BPA detection. Compared with native Tyr or a traditional surface-adsorbed structure of Tyr on Cu–TCPP nanofilms, Tyr@Cu–TCPP retained superior enzymatic activity when exposed to elevated temperatures and extreme acidity or basicity, and the sensor exhibited significantly enhanced thermal and long-term storage stability as well as acid/base tolerance. In this study, the fabrication of an electrochemical biosensor relying on an enzyme assembled between two-dimensional MOF nanomaterial layers has been reported for the first time. This method improved the activity and stability of the biosensor, and therefore provided a promising strategy for promoting the application of enzyme biosensors in harsh detection environments, discovering and controlling environmental pollutants and food hazards [120].

Enzyme-based electrochemical biosensors usually employ chemically immobilized enzyme electrodes. However, limitations of renewability and the possibility of enzyme activity losses are often encountered as drawbacks. A study reported the immobilization of Tyr on the surface of *Escherichia coli* (BL21) cells [121]. The engineered *E. coli* cells displaying Tyr were directly adsorbed on a bare glassy-carbon electrode (GCE) to construct a biosensor for BPA detection. A linear relationship was observed between the concentration of BPA and the current peak. The accuracy of BPA detection using this biosensor was comparable to that of HPLC. The biosensor exhibited a linear relationship in the concentration range of 0.01–100 nM BPA. The detection limit was 0.01 nM, which is lower than that of other chemically modified tyrosinase-based biosensors [121].

Most of the enzyme biosensors for BPA analysis are employing tyrosinase. Brett and co-workers reported the use of xanthine oxidase (XOx) for a BPA biosensor based on the inhibition of XOx activity in the presence of BPA. This strategy based on inhibition is usually beneficial in avoiding that certain chemical compounds present can act as interferents, such as when these compounds also serve as substrates or inhibitors binding to the active site of the biorecognition enzyme, or when they bind to other regions of the immobilized enzyme and cause changes in its active site. For the construction of the sensor, XOx was crosslinked with glutaraldehyde on GCE with hypoxanthine as enzyme substrate, and BPA was determined using amperometry. The biosensor exhibited a low detection limit and enabled successful detection of BPA in water samples [122].

#### 4.5.3. Formaldehyde

In spite the fact that formaldehyde can be found naturally in many animal and plant-based products as an intermediate metabolic product, cases related to food safety are well known, in which formaldehyde is illegally added to various foods, especially to seafood and meat products, in order to extend the shelf life [123]. A long-term exposure to formaldehyde may result in some serious and chronic health problems such as nausea, headache, chest pain, and even death, so that testing for formaldehyde in food can be essential for assuring the safety of food. Biosensors based on enzymes immobilized on SPCE for formaldehyde detection were developed [124]. The authors fabricated a formaldehyde dehydrogenase-based biosensor strip to detect formaldehyde using CV, and compared the results obtained with the sensor to those of an optical enzyme sensor with an α-Fe_2_O_3_/indium-tin oxide bioelectrode. Both the electrochemical and the optical biosensor showed high sensitivity and low detection limits, yet the electrochemical one offered certain additional advantages. It required smaller amounts of sample, it possessed on-site detection portability, and it gave lower RSD values (<2%) for formaldehyde detection in real samples [124]. A membrane-based potentiometric biosensor was fabricated from poly(n-butyl acrylate-co-*N*-acryloxysuccinimide) as both enzyme supporting matrix and pH-sensitive transducer together with alcohol oxidase as bioelement. A linear range for formaldehyde detection from 0.5 to 220 mM together with a response time within 8 s was reported for this biosensor [125]. Table 7 shows a list of biosensors recently developed for the detection and quantitation of various food contaminants.

## 5. Improvement Strategies

### 5.1. Enzyme Engineering

Enzymes from wild-type organisms are predominantly used in the fabrication of biosensor prototypes for food applications, such as *A. niger* glucose oxidase, which was shown to have good selectivity for glucose detection. Nevertheless, these wild-type enzymes can also show some drawbacks including low stability, poor selectivity, or low yields when produced by their original source organisms. Enzyme engineering approaches such as site-directed mutagenesis and directed evolution as well as the fusion of genes to create chimeric proteins with new properties are some strategies used to improve the performance of enzymes for biosensor applications. Site-directed mutagenesis was used to enhance the catalytic activity of pyranose oxidase (POx) from *T. multicolor* for certain substrates. To this end, threonine at position 169 was replaced by glycine, alanine, or serine. Using oxygen as electron acceptor, the variant T169G was equally active with D-glucose and D-galactose, whereas the wild-type recombinant POx only showed 5.2% relative activity with the latter substrate compared to glucose. All the mutations introduced into POx not only showed an improved catalytic response for galactose but also resulted in lower detection limits [34]. Sode’s group reported an engineering approach, whereby they introduced an amino acid on the surface of *An*GOx that was subsequently used as a binding site for a redox mediator, amine-reactive phenazine ethosulfate (arPES). This enabled an electron transfer from FAD in the active site of *An*GOx to an electrode via the tethered redox mediator, and hence a quasi-direct electron transfer (quasi-DET). This approached was based on the 3D structure of the enzyme to select a suitable position for the single amino acid exchange, the introduce of a lysine residue at position 489, to which arPES was then covalently attached [126].

An engineering approach was also carried out with fructose dehydrogenase (FDH), aiming at the improvement of the performance of a DET-type fructose biosensor, by trimming the N-terminus of the enzyme to reduce its size and by introducing a site-directed mutation to decrease the overpotential. The variant M450QΔ1cFDH was constructed by removing 143 amino acid residues involving heme 1c and replacing methionine at position 450 as the sixth axial ligand of heme 2c with glutamine [127]. The DET-type bioelectrocatalytic activity of M450QΔ1cFDH was higher than that of recombinant wild-type FDH. Furthermore, the redox potential was shifted to a more negative value and the surface concentration of the variant on the electrode can be increased since the size of the enzyme was reduced [61]. 

A protein engineering approach for enhancing single biocatalysts does not always yield the expected positive results. Some favourable variants may pose challenges in terms of their expression yields in the host expression system and during protein purification. Moreover, they may exhibit instability when exposed to extrinsic environments and complex matrices, which are essential considerations for many biosensor architectures. In such situations, it may be more advantageous to infer the protein sequence of such an enzyme from an earlier point in its evolutionary history. This can be achieved through convenient computational methods such as the ancestral sequence reconstruction approach. This method allows examination of numerous sequences, potentially leading to the design of new enzymes with improved properties including improved thermostability and specificity [128]. Furthermore, these new sequences may serve as a convenient, stable starting point for subsequent mutational adjustments.

The creation of chimeric proteins comprises the fusion of two or more genes (or parts thereof) that originally coded for separate proteins, thereby creating a single engineered polypeptide with functional properties derived from the original proteins. One of the advantages of using fusion proteins over co-immobilized enzymes is that the resulting protein molecules show a fixed molecular ratio of the individually immobilized enzymes. Sode and co-workers used this approach to create a fusion protein of FAD-dependent glucose dehydrogenase from *Aspergillus flavus* (*Af*GDH) and a heme-containg electron transfer domain of cellobiose dehydrogenase (CDH) from *Phanerochaete chrysosporium* (Pcyb). In order to improve the slow intramolecular electron transfer (IET) rate from the FAD to the heme, Lys substitutions at E324 or N408 were introduced in order to increase the positive charge at the rim of the interdomain region. The IET ability of Pcyb-*Af*GDH increased drastically by introducing these mutations without a loss of the catalytic efficiency. Furthermore, the abilities of engineered Pcyb-*Af*GDH to perform DET to the electrode increased nine-fold for the E324K variant and 15-fold for the N408K variant compared to the wild-type fusion protein [129]. Another study, in which a *Glomerella cingulata* GDH and a *Neurospora crassa* CDH cytochrome domain were merged, thoroughly examined the edge-to-edge distance and charges in the region of the domain interaction region and also showed improved IET as well as DET [130].

### 5.2. Nanomaterials

The application of nanomaterials has clearly shown a number of benefits for biosensor construction in past studies, particularly with regard to the performance of the sensor, such as an increase in the sensitivity, a lowering of the detection limits or enhancement of the biosensor’s stability. Recent developments of electrochemical enzyme biosensors involve the use of working electrodes modified with different nanomaterials. Various types of nanostructures can be synthesized from nanomaterials such as nanoparticles (NPs), nanotubes, nanorods, and hierarchical nanostructures. A variety of nanomaterials, including noble metal nanoparticles, carbon nanotubes [131], and graphene oxide (GO) have been employed to form an appropriate nanostructure layer for better GOx immobilization [39]. Noble metal nanoparticles, e.g., gold, silver and platinum nanoparticles, have been investigated in detail since they can exhibit outstanding properties. One of the advantages of incorporating nanoparticles during the immobilization step of an enzyme is to provide a large surface area for enzymes to be immobilized onto the electrode, thus producing high enzyme loading and high catalytic efficiency. The platinum nanoparticulate on a poly(acryl acid)-modified SPCE electrode resulted in a good environment for the immobilization of GOx, with optimum values of 30 mg·mL^−1^ for enzyme loading and 1 h for the immobilization of GOx [107]. Other studies reporting the application of AuNPs [27,80], PtNPs [37], and AgNPs [73] confirmed these advantageous properties as well. A microelectrode glucose biosensor based on a three-dimensional hybrid nanoporous platinum/GO nanostructure was developed [39]. The 3D hybrid nanostructure was fabricated by a two-step modification method. First, foamy nanoporous structures were prepared by simple electrochemical etching on the surface of a platinum electrode with a 0.5 mm diameter. Then GO was electrochemically deposited on the nanoporous structure to form the 3D nanostructure. The nanoporous structure provided a large number of active sites for effectively capturing GOx molecules in close contact with GO, thus providing an efficient DET process for GOx without any electron mediator. The role of the nanoparticle size on the thermodynamics and kinetics of biocatalytic processes catalysed by oxidoreductases was previously discussed [132]. Gorton and co-workers investigated the effect of the shape of AuNPs on the catalytic current of immobilized FDH. They found that triangular NPs showed more interaction between the enzyme and the NPs, which occurred at the edge of the triangles and thereby contributed more to the total catalytic current, whereas the number of enzyme molecules interacting with NPs was reduced for spherical shapes. The shape of the NPs had no effect on the catalytic constant though (*k*_cat_) [133].

Carbon nanotubes (CNTs) are hollow carbon structures with a single (single-walled, SWCNTs) or several walls (multi-walled, MWCNTs). They have a diameter in the nm-range and show a cylinder-shape, which together provide unique properties and offer promises for a wide range of biosensor applications [134]. The number of walls or innertubes of CNTs plays an important role in supporting the electron transfer (ET) on the electrode. An increase in wall numbers of CNTs will also increase the thickness of the tubes, and therefore, increase the ET distance, which in turn decreases its efficiency. The use of CNTs with 2–3 walls in a glucose biosensor was suggested to provide an optimal balance between these two effects. This also explains the observed two times higher sensitivity of two-walled CNTs compared with single-walled CNTs and CNTs with up to four walls [131]. Other recent studies reported the construction of various biosensors based on CNTs as well [53,104,108]. Surface functionalization of CNTs with carboxyl groups is an attractive way for increasing hydrophilicity and electronic stability of CNTs. Carboxylate groups were introduced to both GO and MWCNT, which was beneficial for fixing GlOx by EDC/NHS coupling. Both GO-COOH and MWCNT-COOH provided excellent catalytic properties and a large surface area for enzyme immobilisation [94]. 

Nanomaterials with a core and shell structure, known as Core@Shell Nanomaterials (CSNs), have gained significant interest in biosensing because of their versatile properties that can be achieved by controlling the core or shell materials. These are made up of a core layer and an outer shell layer of different materials at the nanoscale, which exhibit complementary properties [135,136]. The AuNRs@MS-doped AChE biosensor exhibited a significantly improved sensitivity for OPs detection compared with the bare AuNRs doped as well as the undoped counterparts [113,114]. The architectural design of the CSNs has created additional pathways for electronic signals that are generated by the immobilized DAO in the biosensing system. This has been achieved through the combination of highly conductive polyaniline (PANI) and ZnO nanoparticles [105]. 

### 5.3. Polymers

Polymers are critical materials within the development of biosensors. They are simple to handle, their chemical and physical properties can be custom-made as required, and they can affect the performance of a biosensor significantly [137]. Many of the polymers used in biosensor applications are conductive polymers, which are usually used as coating or encapsulating materials on the electrode surface [138,139]. Meanwhile, non-conductive polymers are used as well, e.g., to immobilize specific receptors on the biosensor device [140]. A variety of conducting polymers including polyacetylene, polypyrrole, polyaniline (PANI), and polythiophene have been extensively used in the design of advanced biosensors. The selection of the polymer matrix, which offers a stable environment for the enzymes and provides appropriate specific functional groups, is a fundamental aspect in biosensor design. Polymers can be used either alone or in combination together for the fabrication of a biosensor. One of the most extensively studied polymers, polypyrrole, has been shown to contribute to the selectivity and stability of a biosensor [71,106]. A recent study also confirmed the benefits of using a combination of a highly conductive PANI matrix together with ZnO nanoparticles for enhancing the electrochemical response of a biosensor, resulting in the highly sensitive detection of histamine of 28.57 μ·mM^−1^·cm^−2^ [105].

The efficiency of using PANI together with polypyrrole deposited by enzymatic polymerization on the surface of graphite rods was evaluated in another study [21]. The electrode used was initially pre-modified by electrochemically synthesizing dentritic gold nanostructures and subsequent drop-casting of GOx for glucose sensing. Enzymatically formed polypyrrole was found to be more suitable for the modification of the working electrode, giving 1.35-times higher sensitivity and a 2.6-times lower LOD than when using PANI. Another polymer studied is poly(3,4-ethylenedioxythiophene) (PEDOT), which is derived from polythiophene. This polymer features remarkable thermal stability, good film-forming properties, a low oxidation potential, excellent transparency, and tuneable electrical conductivity in the doped state [141]. This was also confirmed in a later study, which reported that the use of modified PEDOT, poly(3,4-ethylenedioxythiophene):4-sulfocalix [4] arene (PEDOT:SCX) in combination with a metal carbide-based layered material and GOx exhibited good electrochemical activity through the redox peak of FAD at the formal potential of −0.435 V applied for the electrochemical detection of glucose in fruit juice [25]. The combined use of PEDOT and polypyrrole was also reported in the construction of a lactose biosensor [51]. The sensor was based on a copolymer of polypyrrole and poly(3,4-ethylene-dioxythiophene) synthesized by electro-polymerization together with the bienzymatic system β-galactosidase and galactose oxidase. This film-modified enzyme electrode showed several advantages including a wide linear range together with a low LOD, good temperature stability, a short response time, a strong interaction between enzyme and substrate (low K_m_ value), which all together allowed for the rapid detection of lactose in milk with different fat content. The addition of PEDOT to the polypyrrole structure improved the entrapment of enzymes due to the molecular structure of PEDOT, and improved the analytical performance of the electrochemical biosensor.

Another type of polymer commonly used in the fabrication of electrochemical devices are redox polymers. Redox polymer consists of a non-conductive backbone with redox active pendant groups attached to it. These polymers can act as an electron shuttle via self-exchange-based conduction [142]. Several reports already evaluated the benefits of using redox polymer for the immobilizing of both the enzyme and the mediator at the enzyme surface, showing an improvement of the electron mediation between enzyme and electrode, as well as a minimisation of diffusion limitations [143,144,145]. Erden and co-workers (2019) presented the advantageous use of polyvinyl ferrocene (PVF) in combination with graphene oxide. Whereas an increase in the graphene oxide concentration in the construction of the sensor did not affect the catalytic current significantly, increasing PVF concentration up to 2 mg mL^−1^ improved the biosensor response. Further increases in the PVF concentrations caused a slight decrease in the response though, probably due to lower diffusion rates of the substrate in the thicker PVF films [99].

## 6. Conclusions—Challenges and Outlooks

The use of various analytical devices and especially biosensors can have major implications in the field of food quality and safety, since they can provide a rapid, sensitive, and continuous measurement for food process monitoring. In this review, we discussed different types of electrochemical enzyme-based biosensors in view of potential applications in the food and beverage industries. Among several types of electrochemical transduction systems described, amperometric biosensors are predominantly used for the development of biosensors aiming at food applications. One of the advantages of amperometric biosensors is their generally high sensitivity when compared to sensors based on other transduction systems. The intrinsic simplicity of the amperometric transducer lends itself to the construction of low-cost portable devices for broad applications, especially for the food industry. Amperometry (together with voltammetry) is thus undoubtedly the most suitable electrochemical transducer choice, showing not only high sensitivity but also good working responses over a wide range of analyte concentrations, which is essential for analyte detection in food. In contrast to health applications, biosensors for food applications need to operate over a wide concentration range so that it can be used on a broad variety of different foods and food qualities. A limitation of employing electrochemical measurements is the regeneration between individual measurements using electrodes made of inert metals or carbon. However, the introduction of single-use or disposable screen-printed electrodes, which offer cost efficiency together with the possibility of mass-manufacturing and miniaturization, can overcome this problem [146]. Development of a microfluidic platform is a promising way to achieve miniaturization for a portable yet dependable method of analyzing food samples in the field. Although most studies on microfluidic devices have focused on their application in pharmaceutical and clinical settings due to their small reaction volume for sensing, they are also ideal for food analyte determination.

Enzyme biosensors are very promising when it comes to the development of new biosensors or improvement of existing ones, mainly because of their excellent specificity and high catalytic efficiency. Nevertheless, enzyme-based biosensors may also show some disadvantages, such as loss of enzyme activity due to its interactions with the electrode surface, which can result in a biosensor’s lifespan of only 2–4 weeks. Therefore, strategies to improve the properties of the enzyme itself and the analytical performance of the biosensor are clearly needed. In spite of generally showing high specificity, some enzymes show limitations in the analysis of certain analytes. For example, XOx catalyses both the oxidation of hypoxanthine to xanthine and xanthine to uric acid. Thus, all currently described XOx biosensors, based on uric acid as well as hydrogen peroxide or oxygen consumption measurements, offer the total concentration of hypoxanthine and xanthine present in the samples. Possible ways of mitigating these problems could be offered by enzymes engineering, by which the specificity of an enzyme is tailored towards desired analytes, or the reactivity with other analytes is abolished. Enzyme engineering approaches have also shown impressive results when it comes to the stabilisation of enzymes. Recent advances in computational approaches for enzyme engineering have facilitated this technique and its application significantly. Many studies aiming at the construction of a biosensor focused primarily on the ‘hardware’ of the sensor, i.e., the electrode material, nanomaterials, polymers, and various combinations thereof. Interestingly, enzyme engineering or even screening for novel enzymes in nature has been applied to a much lesser extent when it comes to the design of novel biosensors. It is recommended that material scientists work more closely with enzyme engineers/enzymologists on the development of improved or novel biosensors that then show superior analytical performance. A future trend of biosensors in food and beverage applications will be the development of smart and portable sensors for detection of complex matrices in the food and beverage industry. The emerging biosensor market needs commercialisation of more, better, and novel biosensors as well as offering opportunities to engage with food industries and regulatory institutions responsible for the monitoring of food quality and safety.

## Figures and Tables

**Figure 1 foods-12-03355-f001:**
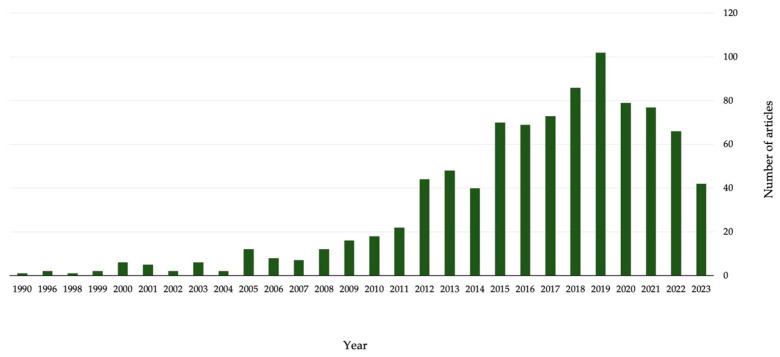
Number of articles related to the development of food and beverage electrochemical biosensors by year. A bibliometric study was conducted using the PubMed database.

**Figure 3 foods-12-03355-f003:**
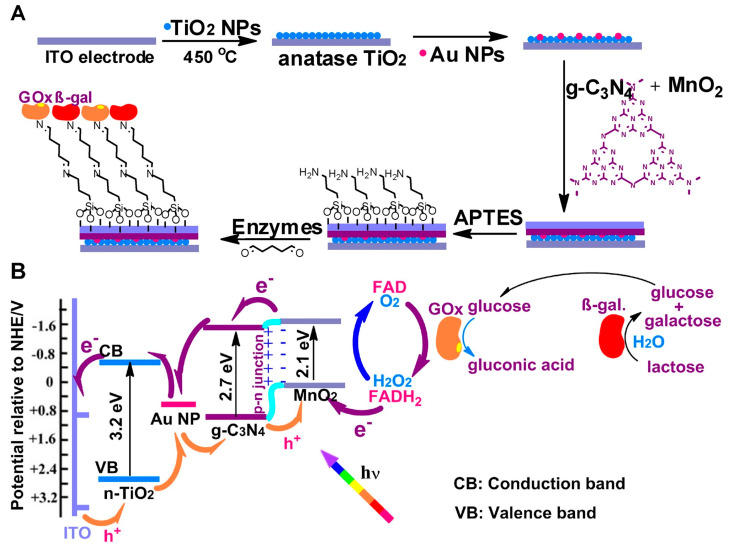
A photoelectrochemical glucose and lactose biosensor. (**A**) Fabrication steps, and (**B**) Biosensing mechanism for glucose, and lactose. Reprinted with permission from Ref. [55]. Copyright 2023, Elsevier.

**Figure 4 foods-12-03355-f004:**
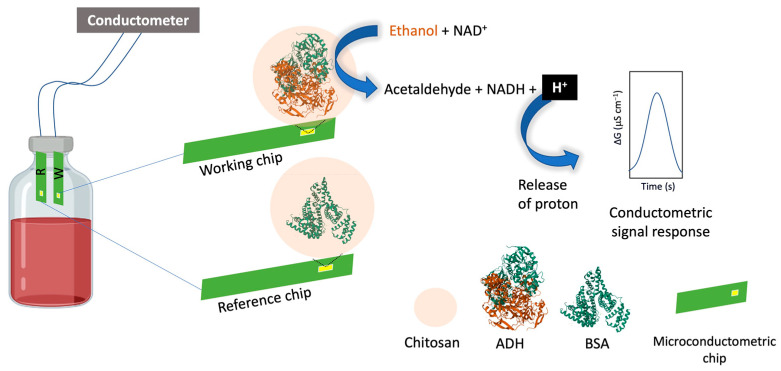
A schematic illustration of a conductometric biosensor for detection of ethanol using a microconductometric electrode.

**Figure 5 foods-12-03355-f005:**
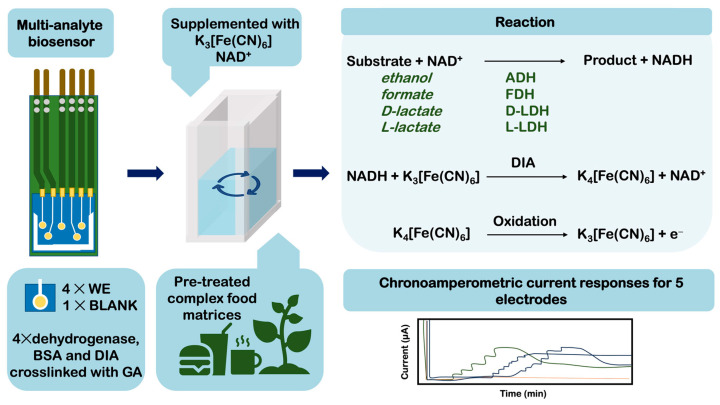
A schematic illustration of the simultaneous detection of three organic acids (formate, D/L -lactate) and ethanol. The illustration was inspired by Figure 2 and Figure 3 of the study [84] and enzymatic reactions presented in the earlier study [85].

**Figure 6 foods-12-03355-f006:**
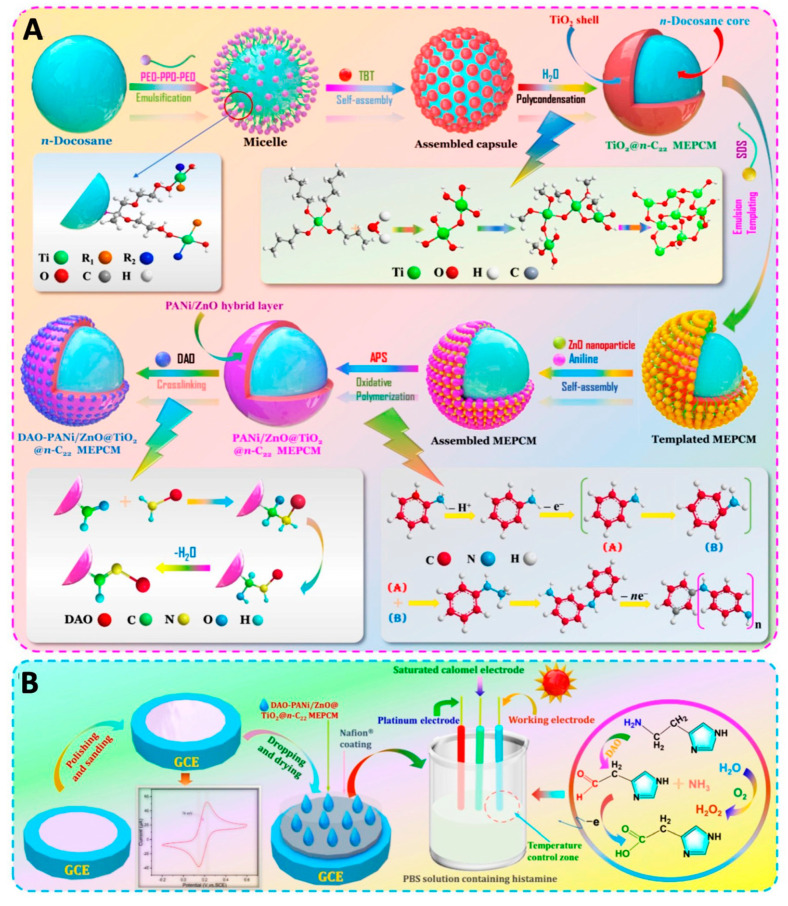
DAO-based smart biosensor for histamine detection. (**A**) The fabrication steps and reaction mechanism. (**B**) Overall sensing mechanism of the biosensor. Reprinted with permission from Ref. [105]. Copyright 2023, Elsevier.

**Figure 7 foods-12-03355-f007:**
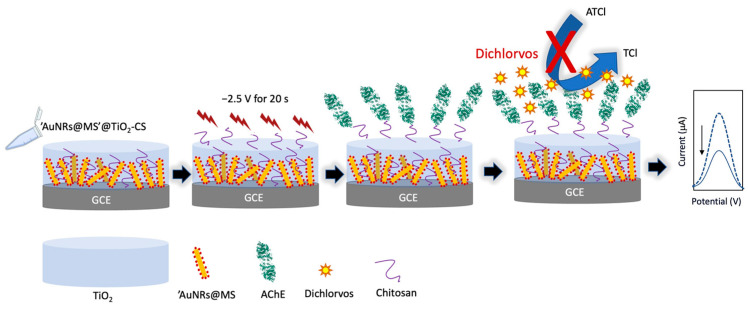
A schematic illustration of amperometric biosensor for detection of dichlorvos and fenthion based on Ref [113].

**Table 1 foods-12-03355-t001:** Different types of transducers used in biosensors.

	Electrochemical	Optical	Mass-Based
Working principle	Detection of the potential/gradient of oxidation and reduction reactions from enzymes and metabolites.	Chemical or biological reactions produce light signals (visible, ultraviolet and infrared) that are measured by a transducer and converted into data for analysis	Production of electrical signals based on applied mechanical force
Advantages	User-friendly Miniaturization Fast detection Low detection limit	High sensitivity and selectivity No electrical interference	Simplicity No optical interference Stable output
Drawbacks	Unstable current and voltage Less selectivity Limited shelf life	Bulky instruments Requirement of sample pre-treatment	Low sensitivity Interference induces by nonspecific binding

**Table 2 foods-12-03355-t002:** Application of enzyme-based biosensors in glucose detection.

Analytes	Electrode	Enzymes	Transducer	Sensitivity	Detection Range	LOD	Food Matrices	Ref.
Glucose	Nafion/MnO_2_-GNR/SPCE	GOx	Amp	56.32 μA mM^−1^ cm^−2^	0.1–1.4 mmol L^−1^	0.05 mM	Honey	[28]
Glucose	PEDOT/PAA/GOx PEDOT/AA/GOx	GOx	Amp	2.74 × 10^−4^ A M^−1^ 2.57 × 10^−4^ A M^−1^	0.96–30 mM 1.86–30 mM	0.29 mM 0.56 mM	Grape juice, honey	[26]
Glucose	Poly(2,2-bithiophene)/Pt disk	GOx	Amp	1.5 × 10^−3^ A mM^−1^	0.09–5.20 mM	30 μM	Pear, apricot, and peach fruit juices	[36]
	Poly(4,4′-bithiophene derivative/Pt disk			3.4 × 10^−4^ A mM^−1^	0.15–5.20 mM	50 μM		
Glucose Galactose	Os polymer/graphite rod	POx	Amp	n.d	0.1–15 mM 0.1–10 mM	8.5 μM 3.2 μM	n.d	[34]
Glucose	PtNPs-poly(Azure-A)-aSPCE	GOx	Amp	42.7 μA mM^−1^ cm^−2^	20 μM–2.3 mM	7.6 μM	Commercial orange, pineapple, and peach juices	[37]
Glucose	MWCNTs/Nafion/GCE	GOx	Amp	23.3 μA mM^−1^ cm^−2^	50 μM–1 mM	0.58 μM	Honey	[38]
				32.4 μA mM^−1^ cm^−2^	1–3 mM	4.94 μM		
Glucose Alcohol	CF(Hemin-AuNPs)/graphite rod	GOx	Amp	909.5 A M^−1^ m^−2^	0.1–0.9 mM	0.05 mM	Grape must and wine	[30]
		AOx		4089 A M^−1^ m^−2^	0.01–0.15 mM	0.005 mM		
Glucose	NPPt/GO/Nafion	GOx	Amp	11.64 μA.L.mmol^−1^ cm^−2^	0.1–4 mmol L^−1^ 4–20.0 mmol L^−1^	13 μM	Tomato, cucumber	[39]
Glucose	PEDOT:SCX/MXene/GOx/GCE	GOx	Amp	n.d	0.5–8 mM	0.0225 mM	Fruit juice	[25]
Glucose	Ppy/GOx/DGNs/ Graphite rod	GOx	Amp	59.4 μA mM^−1^ cm^−2^	0.1–19.9 mmol L^−1^	0.070 mM	Wine, coconut milk, almond milk, apple juice, mandarin juice	[21]
	PANI/GOx/DGNs/ Graphite rod			43.9 μA mM^−1^ cm^−2^	0.3–19.9 mmol L^−1^	0.18 mM		
Glucose	AuNPs/PENDI/PGE	GOx	Amp	0.172 μA mM^−1^ cm^−2^	0.0009–0.33 mM	0.0407 mM	Dextrose solution, orange juice	[27]

LOD, limit of detection; GOx, glucose oxidase; POx, pyranose oxidase; AOx, alcohol oxidase; Amp, amperometry; SPCE, Screen-Printed Carbon Electrode; PEDOT, poly(3,4-ethylenedioxythiophene); polyacrylic acid (PAA); anthranilic acid (AA); GNR, graphene nanoribbons; PtNPs, platinum nanoparticles; MWCNTs, multi-walled carbon nanotubes; CF, carbon microfibers; AuNPs, gold nanoparticles; NPPt, nanoporous platinum; GO, graphene oxide; SCX, 4-sulfocalix; DGNs, dendritic gold nanostructures; PANI, polyaniline; Ppy, polypyrrole; PENDI, poly-*N*,*N*′-bis(2-hexyl)-2,6-(3,4 ethylenedioxythiophene)-1,4,5,8-naphthalenimide; PGE, pencil graphite electrode; n.d, not determined.

**Table 3 foods-12-03355-t003:** Application of enzyme-based biosensors in other saccharides detection.

Analytes	Electrode	Enzymes	Transducer	Sensitivity	Detection Range	LOD	Food Matrices	Ref.
Fructose	4-MPh/h-PG/polycrystalline Au electrodes	FDH	Amp	175 μA mM^−1^ cm^−2^	0.05–5 mM	0.3 μM	Honey, tomato juice, apple juice, pineapple juice, energy drinks	[60]
Fructose	Au microdisk electrode	FDH	Amp	200 μA mM^−1^ cm^−2^	up to 2 mM	n.d	Fruit juice, carbonated drinks, honey	[61]
Sucrose	Chitosan/planar Au electrode	Invertase mutarotase GDH	Amp	0.65 nA μM^−1^	10–1200 μM	8.4 μM	Green coffee beans	[45]
Sucrose	CuNPs-MFC-IGT/AuSPE	Invertase GOx	Amp	3.7 μA M^−1^	0.01 nM–100 μM	0.01 nM	Sweetened tea beverages	[44]
Sucrose	PEI/GA/ silicalite-modified stainless steel electrodes	Invertase mutarotase GOx	Cond	n.d	0.0035–4 mM	3.5 μM	Orange nectar, orange juice, apple juice	[29]
Maltose	Sol-gel-MWCNTs/PVC tube	α-1,4-*glucosidase* GOx	Amp	29.15 μA mM^−1^ cm^−2^	0.5–5 mM	2.4 × 10^−2^ mM	n.d	[49]
Maltose	GDH/Os polymer/Graphite rod	GDH	Amp	1.7 μA mM^−1^ cm^−2^	0.5–15 mM	0.45 mM	n.d	[50]
Lactose	Poly(Pyrrole-co-EDOT)/Pt disc electrode	β-gal GalOx	CV	1.08 A M^−1^ cm^−2^	0.198–2.301 mM	1.4 × 10^−5^ M	Whole, low-fat, skimmed milk	[51]
Lactose Glucose	GOx-β-Gal/Au NPs-graphitic C_3_N_4_-MnO_2_-TiO_2_/ITO	β-gal GOx	Photo-electrochemical	1.66 μA mM^−1^ cm^−2^ (lactose)	0.008–2.50 mM (lactose)	0.23 μM (lactose)	n.d	[55]
				1.54 μA mM^−1^ cm^−2^ (glucose)	0.004–1.75 mM (glucose)	0.12 μM (glucose)		
Lactose	β-gal/MWCNTs/ carbon paste electrode	β-gal	Amp	1.06 μA mmol^−1^L cm^−2^	up to 0.025 mM	0.15 mM	Skimmed milk	[53]
Lactose	Enzyme nanoparticles/Au-wire electrode	β-gal GOx	CV	n.d	1–10 mg mL^−1^	1 mg mL^−1^	Processed milk	[56]
Lactose	Chitosan/enzyme/GCE	β-gal GOx	Pot	9.41 × 10^−4^ C cm^−2^ mM^−1^	5.83 × 10^−3^ to 1.65 × 10^−2^ M	1.38 mM	Whey permeates, milk protein isolates	[62]
Lactose	Poly (meta-phenylenediamine)/Pt disk electrode	β-gal mutarotase GOx	Amp	n.d	0.01–1.25 mM	0.005 mM	Milk	[63]

LOD, limit of detection; FDH, fructose dehydrogenase; GDH, glucose dehydrogenase; GOx, glucose oxidase; GalOx, galactose oxidase; β-gal, β-galactosidase; Amp, amperometry; Cond, conductometry; CV, cyclic voltammetry; Pot, potentiometry; h-PG, highly porous gold; 4-MPh, 4-mercaptophenol; MFC, microfibrillated cellulose; IGT, Indian gum Tragacanth; AuSPE, gold screen printed electrode; GA, glutaraldehyde; PEI, polyethyleimine; MWCNTs, multi-walled carbon nanotubes; EDOT, 3,4-ethylenedioxythiophene; ITO, indium tin oxide; GCE, glassy carbon electrode; n.d, not determined.

**Table 4 foods-12-03355-t004:** Application of enzyme-based biosensors in alcohol and antioxidants detection.

Analytes	Electrode	Enzymes	Transducer	Sensitivity	Detection Range	LOD	Food Matrices	Ref.
Ethanol	AgNPs/PANI/Graphite epoxy composites	AOx HRP	SWV	6.899 μA L g^−1^	Up to 0.35 g L^−1^	3.48 × 10^−3^ g L^−1^	n.d	[73]
Ethanol	TCBQ-LCPs/SWCNTs	ADH	Amp	0.5188 μA mM^−1^	0.2–13 mM	0.05 mM	Beer, red wine, Chinese liquor	[74]
Ethanol	Graphite/(PDDA-CG/electrode	AOx	Amp	n.d	250–1500 μM	50 μM	White and red wine, whisky, vodka	[75]
Ethanol	PAH/SPE	ADH	Amp	13.45 μA mM^−1^ cm^−2^	0.05–2 mM	20 μM	Commercial beer	[67]
Ethanol	chitosan/interdigitated Au electrodes	ADH	Cond	36.8 μS cm^−1^ (*v*/*v*)^−1^	n.d	1200 ppm (220 mM)	Red wine	[68]
Polyphenols	Ppy/AuNPs/SPCE	Lacc	Amp	n.d	1–250 μM	0.83 μM	Propolis	[71]
Polyphenols	PEDOT/SNGC	Tyr	Amp	2.4 × 10^−4^ μA μM^−1^	10–300 μM	4.33 μM	Beers and wines	[70]
Polyphenols	GNP-MnO_2_/SPCE	Lacc	Amp	455 nA µM^−1^	5–320 μM	1.9 μM	Commercial white & red Wine	[76]
Hydroquinone	AuNPs/GNP/SPCE	Lacc	Amp	0.0029 μA μM^−1^	2–120 μM	1.5 μM	Wine & Blueberry syrup	[77]

LOD, Limit of Detection; AOx, alcohol oxidase; HRP, horseradish peroxidase; ADH, alcohol dehydrogenase; Lacc, laccase; Tyr, tyranosinase; SWV, square wave voltammetry; Amp, amperometry; Cond, conductometry; PANI, polyaniline; TCBQ, 2,3,5,6-tetrachloro-1,4-benzoquinone; LCPs, liquid-crystalline lipidic cubic phases; SWCNTs, single-walled carbon nanotubes; PDDA, poly diallyldimethylammonium chloride; CG, carboxylated graphene; SPCE, screen printed carbon electrode; PAH, poly(allylamine hydrochloride); Ppy, polypyrrole; AuNPs, gold nanoparticles; PEDOT, Poly(3,4-ethylenedioxythiophene); SNGC, Sonogel-Carbon electrode; GNP, graphene nanoplatelets; n.d, not determined.

**Table 6 foods-12-03355-t006:** Application of enzyme-based biosensors in amino acids, biogenic amines, and purine derivatives.

Analytes	Electrode	Enzymes	Transducer	Sensitivity	Detection Range	LOD	Food Matrices	Ref.
L-Lysine	Au electrode	LyOx	Amp	n.d	10–800 μM	10 μM	Milk	[93]
L-Lysine	Pt electrode	LyOx	Pot	n.d	30–1300 μM	0.03 mM	Mozzarella	[92]
L-Glutamate	SPPtE/oxidised Ppy/GA-BSA	GluOx	Amp	18.3 mA M^−1^ cm^−2^	0.005–1 mM	1.8 μM	Stock cube, ketchup, Parmigiano Reggiano cheese	[95]
L-Glutamate	Nafion/carboxylated MWNTs)/ Au-Pt NPs/SPE	GluOx	Amp	n.d	2 μM–16 mM	0.14 μM	Tomatoes	[94]
Tyramine	PVF/GO/SPCE	DAO MAO	Amp	7.99 μA mM^−1^ 11.98 μA mM^−1^	0.012–0.99 μM 0.010–0.99 μM	0.61 μM	Cheese	[99]
Tyramine	AuNPs/CNFs-IL-chitosan/GCE	Tyr	DPV	n.d	10–60 μM	3.16 μM	Wine	[109]
Histamine	BSA/GA/SPCE	DAO HRP	Amp	1.31–1.59 μA mM^−1^	2–20 μg mL^−1^	0.11 μM	Yellowfin tuna fillets	[100]
Histamine	BSA/GA/SPCE	DAO	Amp	3.8 nA L mg^−1^	1–75 mg L^−1^	0.5 mg L^−1^	Mackerel and hake fish	[101]
Histamine	TiO_2_-carboxylated MWCNTs-RU-chitosan/SPCE	DAO MAO	Amp	3.39 μA mM^−1^ 2.20 μA mM^−1^	9.9–1100 µM 56–1100 µM	6.9 µM 36 µM	Fish	[97]
Histamine	PB/ITONPs/SPCE	DAO MAO	Amp	1.84 μA mM^−1^ 0.06 μA mM^−1^	6–690 μM 2–32,000 μM	1.9 μM 2.0 μM	Cheese	[98]
Histamine	GA/[Fe(CN)_6_]^3−^/SPCE	DAO	Amp	8.9 nA L mg^−1^cm^−2^	5–75 mg L^−1^	0.97 mg L^−1^	Tuna and mackerel	[102]
Histamine	LDH/µ-ISE microelectrode	DAO HRP	Pot	n.d	10^−8^–10^−3^ M	<10 nM	n.d	[103]
Histamine	Chitosan-AuNPs/PB/MWCNTs/SPCE	DAO	Amp	1.319 ± 0.055 nA μmol^−1^ L at pH 7.50	2.5–125 μM 125–400 μM	1.81 μM (0.2 ppm)	Fish and shrimp	[104]
Histamine	DAO-PANI/ZnO@TiO2@*n*-C_22_ MEPCM	DAO	DPV	28.57 μA mM^−1^ cm^−2^	n.d	0.473 μM	Milk, Beer, Orange juice	[105]
Xanthine	AuNPs/carboxylated/MWCNTs/SPCE	XOx	CV	2.388 μA cm^−2^ μM^−1^	n.d	1.14 nM	Fish	[108]
Xanthine	PtNPs/FPP/Pt disk electrode	XOx	Amp	1.10 A M^−1^ cm^−2^	0.01–0.1 mM 0.1–1.4 mM	48 nM	Fish	[107]
Hypoxanthine	Ppy-paratoluenesulfonate-enzymes/Pt electrode	XOx Uricase	Amp	n.d	5–5000 µM	5 µM	Fish	[106]

LOD, Limit of Detection; LyOx, lysine oxidase; GluOx, glutamate oxidase; DAO, diamine oxidase; MAO, monoamine oxidase; Tyr, Tyrosinase; HRP, horseradish peroxidase; XOx, xanthine oxidase; Amp, amperometry; DPV, different pulse voltammetry; Pot, potentiometry; CV, cyclic voltammetry; SPPtE, screen printed platinum electrodes; Ppy, polypyrrole; GA, glutaraldehyde; GO, graphene oxide; SPCE, screen printed carbon electrode; PVF, polyvinylferrocene; GCE, glassy carbon electrode; IL, ionic liquid 1-butyl-3-methylimida zolium tetrafluoroborate; AuNPs, gold nanoparticles; CNFs, carbon nanofibers; TiO_2_, titanium dioxide; MWCNTs, multiwalled carbon nanotubes, RU, hexaammineruthenium (III) chloride; PB, Prussian blue; ITONPs, indium tin oxide nanoparticles; LDH, layered double hydroxide; µ-ISE, micro-Ion Selective Electrodes; MEPCM, microencapsulated phase change materials; PANI, polyaniline; FPP, ferrocenyl polycyclosiloxane polymers; n.d, not determined.

**Table 7 foods-12-03355-t007:** Application of enzyme-based biosensor in contaminants detection.

Analytes	Electrode	Enzymes	Transducer	Sensitivity	Detection Range	LOD	Food Matrices	Ref.
Malathion Methyl parathion	DAR/PB-SWCNTs/GCE	AChE	CV	n.d	10^−6^–10^−12^ g L^−1^	3.11 × 10^−4^ ng L^−1^ 1.88 × 10^−4^ ng L^−1^	Tap water, purified water, Chinese cabbage	[111]
Monocrotophos Dimethoate	mesoporous SiNPs/GCE	AChE	CV	n.d	0.001–0.003 mg L^−1^	2.51 × 10^3^ ng L^−1^ 1.5 × 10^3^ ng L^−1^	Soft drinks	[112]
Dichlovos Fenthion	Chitosan/‘AuNRs@ mesoporous SiO_2′_@TiO_2_-chitosan/GCE	AChE	CV and EIS	n.d	0.018–13.6 μM	5.3 nM 1.3 nM	Cabbage	[113]
Dichlorvos	Chitosan/TiO_2_ /GCE	AChE	DPV	n.d	1.13–22,600 nM	0.23 nM	Cabbage	[115]
Phosmet	WO_3_/graphitic-C_3_N_4_/Pencil graphite electrode	AChE	Amp	15 μA nM^−1^ cm^−2^	5–125 nM	3.6 nM	Wheat flour	[117]
Eleven organo-phosphorus pesticides Methomyl	AuNPs/mercaptomethamidophos/mercaptohexanol/GCE	AChE	DPV and EIS	n.d	0.1–1500 ng mL^−1^	19 to 77 ng L^−1^ 81 ng L^−1^	Apple and cabbage	[118]
Paraoxon	Ce/Zr-based MOF/MWCNTs/GCE	AChE	Amp and DPV	n.d	0.01–150 nM	0.004 nM	Spinach, cabbage	[116]
Paraoxon	zeolitic imidazolate framework-8/Methylene blue/ITO	AChE	DPV	n.d	20–4000 ng mL^−1^	1.7 × 10^3^ ng L^−1^	Apple, eggplant	[119]
Bisphenol A (BPA)	XOx/GCE	XOx	Amp	n.d	up to 41 nM	1.0 nM	Mineral water	[122]
Bisphenol A (BPA)	Cu–TCPP	Tyr	DPV	n.d	3.5 nM–18.9 μM	1.2 nM	Milk	[120]
Bisphenol A (BPA)	GCE	Tyr	Amp	n.d	0.00001–0.1 μM	0.01 nM	Commercial canned teas and juices	[121]
Formaldehyde	SPCE	FdDH	CV	352 μA mg^−1^ L cm^−2^	0.01–0.5 mg L^−1^	0.03 mg L^−1^	Corn	[124]
Formaldehyde	pnBA-NAS/pHEMA/ Ag/AgCl screen printed electrode	AOx	Pot	59.23 mV/decade	0.5–220 mM	0.1 mM	Fish	[125]

LOD, Limit of Detection; AChE, acetylcholinesterase; Tyr, tyrosinase; FdDH, formaldehyde dehydrogenase; AOx, alcohol oxidase; Amp, amperometry; DPV, different pulse voltammetry; Pot, potentiometry; CV, cyclic voltammetry; EIS, electrochemical impedance spectroscopy; SWCNTs, single-walled carbon nanotubes; PB, Prussian blue; DAR, Diazo-resin; GCE, glassy carbon electrode; AuNRs, gold nanorods; MWCNTs, multiwalled carbon nanotubes; TCPP, tetrakis(4-carboxyphenyl)porphyrin); ITO, indium tin oxide; pnBA-NAS, poly(n-butylacrylate-co-N-acryloxysuccinimide; pHEMA, poly(2-hydroxyethyl methacrylate); n.d, not determined.

## Data Availability

Data sharing not applicable.

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
