# Peer review of "Recent Advances in Electrochemical Enzyme-Based Biosensors for Food and Beverage Analysis"

_foods, 2023, doi:10.3390/foods12183355_

Round 1

Reviewer 1 Report

The manuscript describes the state of the art involving enzymatic biosensors applied to analytes found in food or beverages. Recent publications on the subject are presented. Some suggestions are presented below to improve the material:

1) In the initial chapters there is a lot of repetition of the advantages of the biosensor, can be integrated.

2) Cite Figure 1 in the text

3) In the item "The Electrochemical Biosensor", I think I could mention only electrochemical transducers and their advantages over other transducers: the possibility of miniaturization, format versatility, among others.

4) The item "Enzyme-Based Biosensors" could be changed by enzymatic immobilization methods. Some immobilization methods especially for food detection could also be discussed. I think the addition of a figure would be interesting in this topic. What are the most innovative methods of immobilization?

5) in the area of ​​nanomaterials, I believe it would be important to discuss the use of

core-shell in enzymatic immobilization

6) I think it would be interesting to put a graph of the number of articles in the area

7) in the tables it would be important to also include the architecture of the biosensors described

8) I believe a topic on system miniaturization would be important, especially disposables. Or the theme could be described in those presented.

Author Response

The manuscript describes the state of the art involving enzymatic biosensors applied to analytes found in food or beverages. Recent publications on the subject are presented. Some suggestions are presented below to improve the material:

  • In the initial chapters, there is a lot of repetition of the advantages of the biosensor, can be integrated.

Has been revised and shortened. Please check “Introduction” section

  • Cite Figure 1 in the text

Cited in line 60-61

  • In the item "The Electrochemical Biosensor", I think I could mention only electrochemical transducers and their advantages over other transducers: possibility of miniaturization, format versatility, among others.

We changed the previous table to a new one. The table now only describes the comparison between electrochemical transducers and other transducers. A short overview of different types of electrochemical transducers is still provided in this section since we think it is necessary to provide the readers with a background about the working principle of each type so that the reader can follow the story well – we assume that many readers will be from the food field who might not be that familiar with biosensors

  • The item "Enzyme-Based Biosensors" could be changed by enzymatic immobilization methods. Some immobilization methods especially for food detection could also be discussed. I think the addition of a figure would be interesting in this topic. What are the most innovative methods of immobilization?

We added a new figure about basic immobilisation methods and some possible modifications with representative examples

  • in the area of ​​nanomaterials, I believe it would be important to discuss the use of core-shell in enzymatic immobilization

We added a new paragraph in “improvement strategies : nanomaterials section” to discuss this topic

  • I think it would be interesting to put a graph of the number of articles in the area

Was done by providing a new figure in introduction section (Figure 2)

  • in the tables it would be important to also include the architecture of the biosensors described

We added that in a new column for all the tables (for all of the target analytes) see Table 2 to 7

  • I believe a topic on system miniaturization would be important, especially disposables. Or the theme could be described in those presented.

It might not be necessary to add a new separate topic about system miniaturization, so we added some information on miniaturization in the “conclusions-challenges and outlook” section.

Reviewer 2 Report 

Please fine the attached file.

Author Response

The authors need to provide the schematic illustrations of the detection process presented in the referenced papers that the authors consider important on each target analyte.

We have provided the schematic illustrations

Round 2

Reviewer 1 Report

The suggestions were added in the manuscript.